# Using Design of Experiments to Optimize a Screening Analytical Methodology Based on Solid-Phase Microextraction/Gas Chromatography for the Determination of Volatile Methylsiloxanes in Water

**DOI:** 10.3390/molecules26113429

**Published:** 2021-06-05

**Authors:** Fábio Bernardo, Providencia González-Hernández, Nuno Ratola, Verónica Pino, Arminda Alves, Vera Homem

**Affiliations:** 1LEPABE—Laboratory for Process Engineering, Environment, Biotechnology and Energy, Faculty of Engineering, University of Porto, Rua Dr. Roberto Frias, 4200-465 Porto, Portugal; fabiobernardo@fe.up.pt (F.B.); nrneto@fe.up.pt (N.R.); aalves@fe.up.pt (A.A.); 2Unidad Departamental de Química Analítica, Universidad de La Laguna (ULL), La Laguna, 38206 Tenerife, Spain; mgonzalh@ull.edu.es (P.G.-H.); veropino@ull.edu.es (V.P.); 3Institute of Tropical Diseases and Public Health, Universidad de La Laguna (ULL), La Laguna, 38206 Tenerife, Spain

**Keywords:** design of experiments, volatile methylsiloxanes, wastewater, surface water, solid-phase microextraction, gas chromatography, validation

## Abstract

Volatile methylsiloxanes (VMSs) constitute a group of compounds used in a great variety of products, particularly personal care products. Due to their massive use, they are continually discharged into wastewater treatment plants and are increasingly being detected in wastewater and in the environment at low concentrations. The aim of this work was to develop and validate a fast and reliable methodology to screen seven VMSs in water samples, by headspace solid-phase microextraction (HS-SPME) followed by gas chromatography with flame ionization detection (GC-FID). The influence of several factors affecting the extraction efficiency was investigated using a design of experiments approach. The main factors were selected (fiber type, sample volume, ionic strength, extraction and desorption time, extraction and desorption temperature) and optimized, employing a central composite design. The optimal conditions were: 65 µm PDMS/Divinylbenzene fiber, 10 mL sample, 19.5% NaCl, 39 min extraction time, 10 min desorption time, and 33 °C and 240 °C as extraction and desorption temperature, respectively. The methodology was successfully validated, showing low detection limits (up to 24 ng/L), good precision (relative standard deviations below 15%), and accuracy ranging from 62% to 104% in wastewater, tap, and river water samples.

## 1. Introduction

Siloxanes are organic compounds with a backbone of alternating atoms of silicon (Si) and oxygen (O), with organic groups (e.g., methyl, ethyl, phenyl, etc.) attached to the silicon atoms [1]. The most significant representatives are the low molecular volatile methylsiloxanes (VMSs), which are commonly incorporated in daily products such as personal care products (PCPs). Considering their chemical structure, they may be classified as linear (lVMSs) or cyclic (cVMSs) compounds [2]. Linear and cyclic methylsiloxanes are usually expressed as Ln and Dn, with “n” representing the number of silicon atoms.

Due to their massive use, VMSs are continuously discharged down-the-drain, reaching wastewater treatment plants (WWTPs). Conventional WWTPs are not prepared to degrade them. Thus, VMSs may accumulate on sewage sludge or be discharged into the environment (watercourses) via final effluent [3,4]. Cyclic siloxanes (specially *D*4, *D*5, *D*6) are usually detected at higher levels than linear, possibly due to their higher production and use in PCPs. *D*5 is usually the predominant compound, reaching up to 135 µg/L in influents [5], 6.6 µg/L in effluents [6], and 0.3 µg/L in surface water samples (river water and lakes) [7].

With the main objective of reducing their concentrations in the environment by controlling their main source (PCPs) and minimizing the down-the-drain emissions, in 2018, the European Commission published a restriction on the use of *D*4 and *D*5 in wash-off PCPs, mentioning that, after 31 January 2020, these VMSs shall not be placed on the market in concentrations greater than or equal to 0.1% by weight [8]. In January 2019, the European Chemicals Agency submitted a proposal in order to extend this restriction to *D*4, *D*5 and *D*6 in leave-on cosmetics and other consumer and professional products (e.g., dry cleaning, waxes and polishes, washing and cleaning products) [9]. These alterations were proposed due to environmental concerns with those compounds, since some of them were classified as PBT/vPvB (persistent, bioaccumulative and toxic/very persistent and very bioaccumulative) substances. It is estimated that around 90% of VMSs present in PCPs are volatilized during use, with the remaining 10% being discharged down-the-drain to WWTPs [4]. Thus, wastewater can be considered one of the most significant sources of environmental contamination, which highlights the importance of monitoring the VMSs concentrations in the discharges and receiving media.

A literature review indicates that most studies regarding the determination of VMSs in water samples are based on gas chromatography-mass spectrometry (GC-MS) analysis [10]. This is a complex equipment, which requires specialized skills to operate, inhibiting its use in most WWTP laboratories for monitoring purposes. Therefore, in this work, a simple and cheaper instrumental analysis based on gas chromatography-flame ionization detection (GC-FID) is proposed, with the aim of being used as screening method for a large number of samples, before using more complex and expensive equipment, such as GC-MS, for confirmation purposes [6]. The critical point in the determination of VMSs in wastewater samples is the extraction method, which usually involves a cleanup and pre-concentration step. These procedures are usually laborious, very time-consuming and use high amounts of solvents. Liquid-liquid extraction is one of the most frequently applied techniques to determine VMSs in wastewater, but it has the problems referred above [10]. Thus, the search for cheaper, more efficient, and environmentally friendly techniques, requiring fewer solvents and with the possibility of automation is advisable, which justifies the option for the use of solid-phase microextraction (SPME). This analytical alternative enables the determination of trace-level compounds in water samples, using a “green” extraction approach. The efficiency of this extraction process depends on several parameters (e.g., fiber type, extraction and desorption time and temperature, ionic strength, etc.), and to achieve the optimal experimental conditions, a design of experiments (DoE) is advisable. This is an alternative to the conventional step-by-step optimization procedure, and it allows to account for possible cross-effects and minimize the number of necessary tests, without losing essential information. In fact, this optimization approach usually requires fewer experimental repetitions and allows for the study of unranked discrete variables, but it has limited suitability for experiments, in which the emphasis is on the ability to extrapolate the results [11]. Therefore, the aim of this study is to develop, optimize using DoE, and validate a solid-phase microextraction followed by GC-FID analysis for the determination of seven VMSs (*D*3, *D*4, *D*5, *D*6, L3, L4, and L5) in aqueous samples.

## 2. Results and Discussion

### 2.1. Optimization of the Extraction Procedure

The optimization of the extraction procedure was performed using a design of experiments (DoE). A two-level fractional factorial design was used as the screening design (SD) in order to understand which variables affect the response. After that, a second order model central composite design (CCD) was used to optimize the extraction methodology.

#### 2.1.1. Screening Design

Several parameters may influence the HS-SPME extraction. Therefore, a screening design considering seven main factors (*X*i) was implemented. Two levels (defined as −1 and +1 for the lower and upper limit, respectively) were set for each of the seven main factors, as presented in Table 1: *X*1—ionic strength (0–20%, *w*/*v*), *X*2—extraction time (5–45 min), *X*3—desorption time (1–10 min), *X*4—extraction temperature (25–80 °C), *X*5—desorption temperature (200–250 °C), *X*6—type of fibers (100 µm polydimethylsiloxane (PDMS), 65 µm PDMS/Divinylbenzene (DVB)) and *X*7—sample volume (5–10 mL). A 2^7–3^ fractional factorial screening design was chosen (resolution IV), leading to a total of 16 experiments, which corresponds to one eighth of the full factorial design with 128 experiments (Table 2). All the extractions were tested using a concentration of 1 µg/L for both VMSs and internal standard (M4Q).

The main results are presented in Figure 1. The model was applied to all compounds and its suitability determined. All responses were adjusted to a quadratic model with a R^2^ > 0.94. The main effects were determined by the F-probability, where F-probability ≤ 0.05 represents a significant effect and if 0.05 < F-probability ≤ 0.1, a relative effect may be considered. Since the main goal was to maximize the response areas for all compounds, a desirability function was applied in order to obtain the best compromise value for all variables.

As presented in Figure 1, all main factors were significant for at least one compound and the extraction temperature (*X*4), fiber type (*X*6), and sample volume (*X*7) were the factors with stronger effects on the response (F-probability < 0.0001). The fiber type is a discrete variable, and therefore to proceed with the model, it had to be defined in advance. A PDMS/DVB fiber was selected since it provided higher responses for all compounds. In this type of fiber, the extraction mechanism is based on adsorption into pores, which is a stronger and more efficient mechanism than absorption, making this fiber more suitable for trace analysis. Analyzing the screening design results, the larger the sample volume, the greater the instrumental response. Therefore, the maximum sample volume tested, 10 mL, was chosen (half of the vial capacity). These results are consistent with two other studies found in the literature that use HS-SPME for the analysis of VMSs in water, although GC-MS was used in both [12,13].

#### 2.1.2. Central Composite Design

After selecting the most important factors using a screening design, those selected parameters were optimized by a central composite design (CCD). Considering that five out of the seven factors were relevant (ionic strength, extraction and desorption time, and extraction and desorption temperature), a total of 26 experimental runs were performed, including six assays in the center of the cubic domain (pattern 00000), resulting in a total of 32 experiments. The CCD factor values and experiments are summarized in Appendix A. The experimental data were fitted to a second-order polynomial equation and the model coefficients were determined based on a least-square regression analysis. The equations and the model suitability were assessed using the ANOVA approach and the main results are presented in Appendix A (only the factors that significantly affect the response are shown). The R^2^ values ranged from 0.80 to 0.97, which indicates a reasonable relationship between the experimental data and the fitted model. Models for most compounds show a F-probability below 0.1, indicating that variations in the response are associated with the models, rather than with the experimental error. However, M4Q and L5 show a higher F-probability, indicating a strong influence of experimental errors, possibly due to some problems related to competition within the fiber pores. Appendix A presents the relevant variables and interactions identified by the Student’s *t*-test, where the values in bold indicate the variables that present significant effect. Figure 2 presents the results from the Student’s *t*-test for the main and quadratic effects. The main factors (*X*1 to *X*5) are the same defined in the screening design. None of the interactions presented Prob > |t| lower than 0.05. As can be observed, the “extraction time” (*X*2) and “extraction temperature” (*X*4) are the variables that significantly affect a larger number of responses, as defined by the low Student’s *t*-test values.

Due to the high number of responses, the desirability function was applied to obtain the best compromise value for all variables, maximizing the response areas (Appendix A). The optimal conditions were: 19.5% NaCl, 39 min extraction time, 10 min desorption time, 33 °C extraction temperature and 240 °C desorption temperature (D = 0.71). The optimization indicates that the addition of NaCl favors the transfer of analytes to the headspace and, consequently, to the fiber, due to the decrease of VMSs solubility in water (although it may also favor the co-extraction of interferents). Extraction time affects the amount of analyte extracted in the fiber coating, and this extraction is maximal when equilibrium is reached, achieving the highest sensitivity. However, in equilibrium extraction, competition might occur at higher concentrations. In this study, the optimal extraction temperature was set at 33 °C. Although the adsorption of fewer volatile compounds is favored by the increase in temperature (due to the improved mass transfer process of VMSs from sample to headspace), the partition coefficient between the coating and the sample decreases with increasing temperature, negatively affecting the adsorption of more volatile analytes. The desorption temperature of 240 °C is in the temperature range supported by the fiber and is high enough to properly desorb previously extracted molecules without damaging the fiber.

Figure 3 presents an example of three-dimensional response surface plot for *D*6. Figure 3A presents the relationship between ionic strength and extraction time, Figure 3B the relationship between extraction time and desorption time, and Figure 3C the relationship between ionic strength and extraction temperature. As can be seen, the combined effect of the addition of salt (19.5% NaCl), together with the increase of desorption time (10 min) and extraction time (to 39 min), maximize the response for this compound.

To prove the applicability of the empirical model, five additional tests with random patterns of the coded levels were performed. Results obtained were in the range of the predicted values by the proposed model and are presented in Appendix A.

### 2.2. Method Validation

Using the optimal conditions, the HS-SPME method performance was evaluated in terms of linearity, method detectability, precision, and accuracy. Table 3, Table 4 and Table 5 present the calculated validation parameters.

Regarding linearity of the response within the range of concentrations presented in Table 3, the correlation factor (R) was higher than 0.995 for all VMSs, except *D*3 and *D*6 and the relative standard deviation of the slope was less than 5%. Two different methods were used to evaluate the limits of detection. For linear siloxanes, the limits of detection (LODs) and quantification (LOQs) were estimated based on a signal-to-noise ratio (S/N) of 3 and 10, respectively. For cyclic siloxanes, they were determined based on 10 lab blanks, i.e., as the concentration of analyte that provides a response equal to the three and ten times their standard deviation, respectively. Nevertheless, these detection limits are still adequate for the intended screening approach using GC-FID.

Intra-day precision (repeatability) was evaluated through relative standard deviation (%RSD) of four replicates at different spiking levels (1 µg/L and 5 µg/L), while inter-day precision (intermediate precision) was assessed by determinations of the same sample, analyzed in different days (three days). In both cases, most relative standard deviation values were below 15%, as presented in Table 4. The mean relative standard deviation (RSD) values were about 12% for intra-day precision and about 15% for the inter-day precision, which is acceptable for this type for HS-SPME analysis.

Wastewater, tap water, and river water were used for accuracy measurements, which expresses the proximity of the obtained analytical result to the expected value. A standard-addition approach was used, by adding 25 ng of siloxanes to the samples and evaluating the resulting percentage of relatively recovery (%R defined as the ratio between the measured and the expected mass after the standard addition). Therefore, the concentrations of VMSs in the sample were calculated by the standard addition quantification method [14]. The matrix effect was previously reported for complex samples as wastewater, resulting in a signal suppression of 12–41% [12]. This was also verified in the present study. Therefore, to overcome this problem, a matrix-matched calibration was implemented for the analysis of wastewater samples. The obtained recoveries are also presented in Table 4. Examples of chromatograms may be seen in Appendix A.

Finally, these parameters of validation were used to calculate the expanded uncertainty associated to final results accordingly to the protocol published by EURACHEM/CITAC Guide [15] and also Konieczka, and Namieśnik (2010) [16]. The global uncertainty was determined by identifying, estimating and combining all sources of uncertainty associated with the result [17,18]. The four main sources of uncertainty considered were: the uncertainty associated to the standards preparation (U1), calibration curve (U2), precision (U3) and accuracy (U4). Figure 4a shows the global uncertainty of *D*5, the compound referred in the literature as the most abundant in water. When concentrations decrease, approaching the limits of detection, the global uncertainty rises significantly. In Figure 4b the variation of the relative weight of each individual source may also be seen for *D*5. For the concentrations between 0.125 and 1 µg/L, U2 accounts for more than 50% of the global uncertainty. The standard preparation (U1) contributes around 10% or less for the global uncertainty. In the case of precision (U3) and accuracy (U4), they are increasingly higher as the upper concentration levels are reached, with their joint contribution reaching up to 70%. The global uncertainty for the other siloxanes follows the same trend as the presented in both figures (data not shown). To ensure a level of confidence of approximately 95%, results are usually expressed as the expanded uncertainty, which is the value of U multiplied by a factor of coverage of 2.

### 2.3. Application of the Developed Method to Environmental Samples

Water samples were analyzed using the developed HS-SPME method. Results of the analyzed waters and respective uncertainty are presented in Table 5.

Siloxanes were not detected in both tap water and river water. In the wastewater sample, cyclic siloxanes presented higher concentrations, reaching up to 0.70 µg/L, which is in accordance with literature values. Linear siloxanes were detected with concentrations up to 0.44 µg/L. L3 and L4 concentrations were higher than those found in literature, while L5 is in accordance with the reported values, reaching up to 0.27 µg/L in influent [12]. The wastewater sample should be further analyzed by GC-MS, following the exact same procedure, in order to confirm the results.

In short, this methodology is suitable for the detection of VMSs in higher concentrations, such as effluents of industries that produce siloxanes or that utilize them in their working process. In addition, this methodology is particularly interesting for aqueous samples with low amounts of dissolved/suspended organic matter.

## 3. Materials and Methods

### 3.1. Materials

Individual standards of each VMSs (L3–L5 and *D*3–*D*6; purity > 97%) and tetrakis(trimethylsilyloxy)silane (M4Q), used as internal standard, were purchased from Sigma-Aldrich (St. Louis, MO, USA). For SPME, two types of fiber, namely 100 µm polydimethylsiloxane (PDMS) and 65 µm PDMS/Divinylbenzene (DVB), were also obtained from Sigma-Aldrich. A Hei-Standard magnetic stirrer with heating from Heidolph Instruments (Schwabach, Germany) was also used in the SPME extraction. Helium (99.999%) used in the GC-FID system, was supplied by Air Liquide (Maia, Portugal).

Individual stock solutions of each siloxane, including the internal standard M4Q, were prepared in *n*-hexane at 1 g/L. From those individual stock solutions, a 5 mg/L mix stock solution containing all the target analytes were prepared in acetone. A diluted M4Q individual stock solution, with a final concentration of 5 mg/L, was also prepared in acetone. From those stock solutions, new mix solutions with 50 μg/L and 500 μg/L were prepared in acetone. Eight calibration standards in acetone, with concentrations of each analyte ranging from 0.125 μg/L to 5 μg/L (internal standard concentration of 3 μg/L) were also prepared. All the solutions were stored protected from light and at −22 °C.

### 3.2. Solid-Phase Microextraction Procedure

The starting point to define the operational variables was the published works that analyzed L3–L4 and *D*3–*D*6 in wastewater, and L2–L5 and *D*3–*D*6 in river water, respectively [12,13].

Ten mL water sample (containing 19.5% NaCl) was placed in a 20 mL screw cap amber vial (75.5 × 22.5 mm) with 18 mm, steel magnetic cap containing polytetrafluoroethylene (PTFE)/butyl septa. A small PTFE-coated stir bar was placed inside the vial and 30 ng of the internal standard (M4Q) was then added. A Supelco SPME holder for manual injection was used. Before the headspace solid-phase microextraction (HS-SPME) analysis, the sample vial was conditioned for 10 min in a bathwater, that was placed on top of the heated magnetic stirrer, at the extraction temperature of 33 ± 5 °C. Then, samples and calibration solutions were extracted with the SPME fiber at 33 °C for 39 min using a constant magnetic agitation rate of 750 rpm. Headspace mode of extraction was selected because VMSs are semi-volatile and due to the possible damage that the fiber could take during wastewater analysis, owing to the sample agitation and particulate matter. Finally, thermal desorption of the analytes was carried out by exposing the fiber in the GC injector port at 240 °C for 10 min.

### 3.3. GC-FID Analysis

The determination of the VMSs was carried out on a GC 2010 Plus gas chromatograph (Shimadzu, Kyoto, Japan). The chromatographic separation of the target compounds was performed on a Low-bleed DB-5MS ultra-inert column (5% phenyl, 95% methyl polysiloxane) fused silica capillary column (J&W Scientific, Folson, CA, USA), 30 m × 0.25 mm I.D., 0.25 μm film thickness. The oven temperature was programmed as follows: 35 °C hold for 5 min, raised at 10 °C/min to 95 °C, then 5 °C/min to 140 °C, and then 35 °C/min to 300 °C (hold for 5.5 min). The total time of analysis was 30 min. Helium was used as a carrier gas at a constant flowrate of 1 mL/min held by electronic flow control. The injector temperature was maintained at 240 °C and the splitless injection mode was used. The detector temperature was set at 300 °C. An SPME glass inlet liner (I.D., 0.75 mm, Shimadzu, Kyoto, Japan) and premium low bleed plasma coated septa (Shimadzu, Kyoto, Japan) were used.

### 3.4. Quality Assurance and Control (QA/QC)

Due to the VMSs ubiquity, analysts in the laboratory did not use personal care products and switched gloves whenever handling different samples. Glassware was subject to a special cleaning and decontamination procedure by pre-rinsing with acetone and distilled water. Non-calibrated material proceeded to further heating at 400 °C, for at least 1 h, in a laboratory muffle furnace. Procedural blanks (extraction of distilled water spiked with internal standard) were analyzed with every extraction batch, to correct sample concentration and identify any background contamination.

### 3.5. Design of Experiments

A design of experiments (DoE) consists of a series of runs that help to investigate the effects of input variables (factors) on an output variable (response). The factors that can influence the response are selected and evaluated. The main goal can be to identify significant factors that affect the response, identify factor interactions, to find optimal factor settings, match a target value or to build a predictive model, using the minimum number of experiments possible [19]. DoE is usually divided in three stages, starting with a screening design (identify the most important factors and reduce the number of factors that will be optimized), followed by a response surface design (construct a design that models a quadratic function of continuous factors, identifying factor settings that optimize the response) and finally the method validation.

Several models can be used for screening (e.g., full factorial and fractional factorial). Factorial designs are used to study the effects of two or more variables on the response. Since the size of full factorial designs increases exponentially with the number of input variables, fractional factorial designs are preferred when dealing with high-dimensional problems. The performed 2^7–3^ fractional factorial screening design (16 experiments, 1/8 of the full factorial design with 128 experiments) corresponds to a resolution IV design, meaning that the main effects are not confounded with other two-factor interactions. Another option would be to apply a 2^7−4^ fractional factorial, that would result in only 8 experiments. However, this design corresponds to a resolution III design, which means it is possible to estimate the main effects, but these may be confounded with two-factor interactions. So, the 2^7–3^ fractional factorial was preferred, even though it meant to conduct double the experiments. After performing the screening design and identifying the significant factors, a response surface methodology (RSM) is applied in order to model a quadratic function of continuous factors, identifying factor settings that optimize the response. Some examples include central composite design (CCD), Plackett–Burman design (PBD), Box–Behnken design (BBD), and Doehlert matrix design. The performed central composite designs consist of two-level full or fractional factorial design, coupled with two-star points (−α, +α) that represent new extreme values (low and high) for each factor in the design that allow the estimation of curvature. It also includes center points in order to measure stability and inherent variability of the model.

Each factor value (*X*_i_) must be transformed into coordinates within a scale with dimensionless values (*x*_i_), which must be proportional to its location in the experimental space. It enables the investigation of variables of different orders of magnitude without influencing the evaluation of the smallest. This means that the models operate on coded dimensionless input values *x*_i_ (−1, +1) instead of actual factor values (*X*_i_), that are converted by the following equation [20] (Equation (1)):(1)$$x_{i}=\frac{X_{i}-X_{0}}{\Delta X}$$
where *X*_0_ refers to the value of variable *i* in the center of the domain (*x_i_* = 0) and Δ*X* refers to the difference of that variable between *x_i_* = +1 and *x_i_* = 0.

After conducting the experiments, the experimental responses must be adjusted to a quadratic model [20,21] (Equation (2)):(2)$$Y=b_{0}+\overset{k}{\underset{i=1}{\sum }}b_{i}x_{i}+\overset{k}{\underset{i=1}{\sum }}b_{ii}x_{i}^{2}+\overset{k}{\underset{j>1}{\sum }}\overset{k}{\underset{i=1}{\sum }}b_{ij}x_{i}x_{j}$$
where *Y* corresponds to the process response, *x_i_* to the codified independent variable, *b_0_* is the interception term, *b_i_* is the influence of the variable *i* in the response, *b_ii_* is the parameter that determines the shape of the curve, and *b_ij_* corresponds to the effect of the interaction among variables *i* and *j*. The quality of the fitted model can be evaluated by applying the analysis of variance (ANOVA), which is used to compare the variations due to the treatment with the variations due to random errors inherent to the measurements of the generated responses. Variations that occur in the response can be attributed to the model and are not due to random errors if the F-probability is less than 0.05 (for 95% confidence level). The relevant variables and interactions can be identified by the Student’s *t*-test. If *t*-probability is smaller than 0.05, the parameter or interaction is considered significant.

When several responses need to be evaluated (in this case, the main goal was to maximize the response areas for all compounds), a desirability function for each individual response is created, transforming each response into a dimensionless individual desirability (*d_i_*) scale, that ranges from *d_i_* = 0 for inacceptable value to *d_i_* = 1 for the single response maximum. The response is considered unacceptable if it is outside the desirable limit [20] (Equation (3)):(3)$$d_{i}^{max}=\{\begin{array}{c} 0\;\;\;\;\;if\;f\left(x\right)<A\\ {\left(\frac{f\left(x\right)-A}{B-A}\right)}^{w}\;if\;A\leq f\left(x\right)\leq B\\ 1\;\;\;\;if\;f\left(x\right)>B \end{array}$$
where *d_i_* is the individual desirability values, *f*(*x*) the response value, *A* is the minimum response value, *B* the maximum response value, and *w* is the weight used to determine the importance (in this work, equal weights were chosen for all responses). It is then possible to calculate the overall desirability (*D*), also ranging from 0 to 1 [21] (Equation (4)):(4)$$D=\sqrt[{n}]{d_{1}d_{2}\ldots d_{m}}$$
where *m* is number of responses studied in the optimization process. The optimized conditions indicate the values of input variables for which the overall desirability is maximal.

In order to generate the experimental matrix for the design of experiments, to evaluate the effects of several parameters and to perform the data analysis, the JMP 14.3 (SAS Institute Inc., Cary, NC, USA) statistical software was used. However, this could be substituted by other similar statistical software.

## 4. Conclusions

This work demonstrates the relevance of experimental design and method validation. Indeed, it goes further, by linking the two components and applying them to the analysis of VMSs in water. The application of design of experiments (DOE) enabled the study of the effect of input variables (factors) on an output variable (response) using the minimum number of experiments possible, thus saving time and resources. The developed HS-SPME technique emerges as an interesting solution to conventional procedures since it combines the potential to eliminate solvent consumption and the concomitant issues of used solvent disposal, with a simple, low-cost, and eco-friendly extraction procedure. In terms of performance parameters, the developed HS-SPME method presents similar results to those obtained by other studies found in the literature that also used HS-SPME for the analysis of VMSs in water samples. The present work also has the advantage of using a simple and cheaper instrumental methodology based on GC-FID, which is more easily replicable in WWTP analysis laboratories, to screen a larger number of samples.

## Figures and Tables

**Figure 1 molecules-26-03429-f001:**
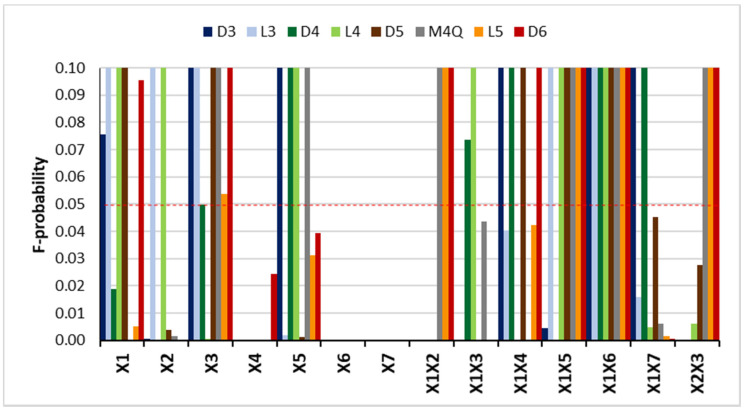
F-probability obtained with screening design.

**Figure 2 molecules-26-03429-f002:**
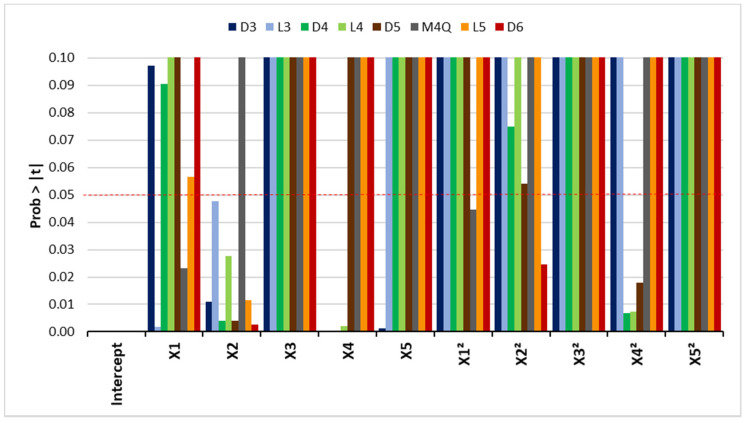
Results from the Student’s *t*-test for the main and quadratic effects.

**Figure 3 molecules-26-03429-f003:**
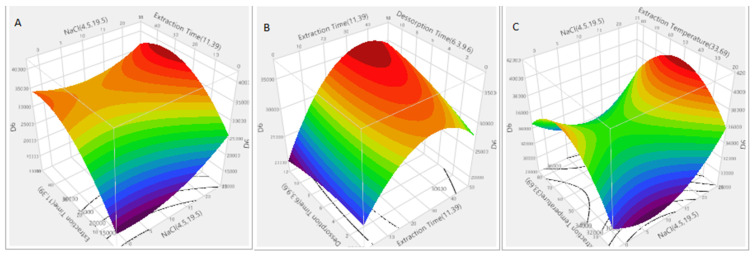
Examples of response surface plots for *D*6: (**A**) effect of extraction time and ionic strength (% NaCl) on peak area; (**B**) effect of extraction and desorption time on peak area; (**C**) effect of extraction temperature and ionic strength (% NaCl) on peak area.

**Figure 4 molecules-26-03429-f004:**
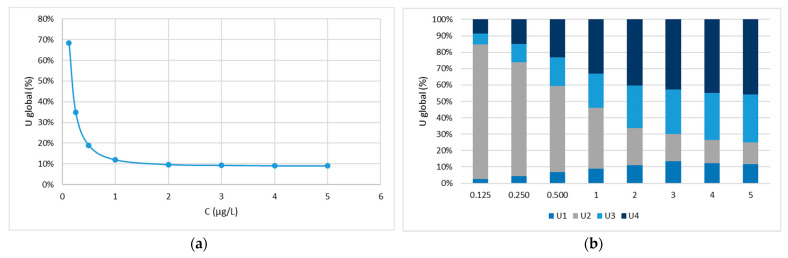
Global uncertainty of *D*5 (**a**) and variation of the relative weight of each individual source of uncertainty for *D*5 (**b**).

**Table 1 molecules-26-03429-t001:** Experimental factors and coded levels for the proposed screening approach.

*i*	Factor	Coded Values (*x_i_*)	
		Low (−1)	High (+1)
1	Ionic strength (% NaCl)	0	20
2	Extraction time (min)	5	45
3	Desorption time (min)	1	10
4	Extraction temperature (°C)	25	80
5	Desorption temperature (°C)	200	250
6	Fiber type	PDMS	PDMS/DVB
7	Sample volume (mL)	5	10

**Table 2 molecules-26-03429-t002:** Proposed screening design.

Run	*X*1 Ionic Strength (% *w*/*v*)	*X*2 Extraction Time (min)	*X*3 Desorption Time (min)	*X*4 Extraction Temperature (°C)	*X*5 Desorption Temperature (°C)	*X*6 Fiber Type	*X*7 Sample Volume (mL)
1	0	5	1	25	200	PDMS	5
2	20	45	10	25	200	PDMS	5
3	0	45	10	25	200	PDMS/DVB	10
4	20	5	1	25	200	PDMS/DVB	10
5	0	45	1	25	250	PDMS	10
6	20	5	10	25	250	PDMS	10
7	0	5	10	25	250	PDMS/DVB	5
8	20	45	1	25	250	PDMS/DVB	5
9	0	5	10	80	200	PDMS	10
10	20	45	1	80	200	PDMS	10
11	0	45	1	80	200	PDMS/DVB	5
12	20	5	10	80	200	PDMS/DVB	5
13	0	45	10	80	250	PDMS	5
14	20	5	1	80	250	PDMS	5
15	0	5	1	80	250	PDMS/DVB	10
16	20	45	10	80	250	PDMS/DVB	10

**Table 3 molecules-26-03429-t003:** Linearity range and limits of detection and quantification for VMSs analysis by HS-SPME/GC-FID.

Analyte	Linearity Range (µg/L) (*n* = 8)	Correlation Factor of the Calibration Curve (R)	Limit of Detection (µg/L)	Limit of Quantification (µg/L)
L3	0.125–5	0.998	0.024	0.080
L4	0.125–5	0.997	0.014	0.047
L5	0.125–5	0.998	0.018	0.061
*D*3	0.125–5	0.992	0.015	0.050
*D*4	0.125–5	0.997	0.015	0.049
*D*5	0.125–5	0.996	0.018	0.059
*D*6	0.125–5	0.993	0.014	0.046

**Table 4 molecules-26-03429-t004:** Precision and accuracy parameters.

Analyte	Intra-Day Precision, *n* = 4 (%RSD)		Inter-Day Precision, *n* = 3 (%RSD)		Accuracy (% Mean Recovery ± SD)		
	1 µg/L	5 µg/L	1 µg/L	5 µg/L	Wastewater	Tap Water	River Water
L3	10	10	12	15	102 ± 3	100 ± 27	104 ± 12
L4	10	12	12	18	79 ± 8	81 ± 4	100 ± 8
L5	10	12	11	19	94 ± 5	88 ± 18	75 ± 16
*D*3	10	14	13	17	84 ± 14	102 ± 26	87 ± 11
*D*4	14	12	14	17	89 ± 5	82 ± 16	101 ± 12
*D*5	13	11	12	18	76 ± 10	62 ± 8	85 ± 13
*D*6	10	11	12	19	93 ± 15	70 ± 10	74 ± 18
Average	11	12	12	17	88 ± 8	84 ± 16	89 ± 13

**Table 5 molecules-26-03429-t005:** VMSs concentrations in real samples ± Global Uncertainty.

Analyte	Wastewater (µg/L)	Tap Water (µg/L)	River Water (µg/L)
L3	0.14 ± 0.23	nd	nd
L4	0.44 ± 0.10	nd	nd
L5	0.27 ± 0.12	nd	nd
*D*3	0.67 ± 0.11	nd	nd
*D*4	0.39 ± 0.16	nd	nd
*D*5	0.34 ± 0.19	nd	nd
*D*6	0.70 ± 0.13	nd	nd

## Data Availability

The data presented in this study is available on request from the corresponding author.

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
