# Peer review of "Using Design of Experiments to Optimize a Screening Analytical Methodology Based on Solid-Phase Microextraction/Gas Chromatography for the Determination of Volatile Methylsiloxanes in Water"

_molecules, 2021, doi:10.3390/molecules26113429_

Round 1
Reviewer 1 Report
The manuscript „Using Design of Experiments to Optimize a Screening Analytical Methodology Based on Solid-Phase Microextraction/Gas Chromatography for the Determination of Volatile Methylsiloxanes” in Molecules is interesting and it could be read by a wide group of scientists worldwide. However, it requires polishing in terms of presentation. It is rather chaotic in the current form.
Keywords: Design of Experiments; Volatile methylsiloxanes; Wastewater
Design of Experiments – why is this a keyword? Keywords need to be targeted so that scientists worldwide can use them for a literature search. "Validation" is a better choice of keyword, however still quite broad. Please add other keywords.
How can the section Material and methods be contained within a subsection of Results and Discussion?? The research has to be conducted first, so that the Results can be described and interpreted in Discussion.
It would be best if there was a schematic drawing shown first, where it will be explained what will be done and how, in chronological order of method steps - e.g., the Ishikawa diagram. Then, add the tools which have been used for the purpose (including explanation and literature backing for their choice). Finally, please present the results obtained and their discussion (and conclude whether they are satisfactory or not). The order needs to be kept for clarity. It is interesting to attach also images of chromatograms.
The information concerning the equipment used and the analytical procedure, including QA/QC, would be best presented in a table.
Conclusion
The Conclusion should summarize the conducted research in the context of already published data (existing knowledge). How does the described analytical procedure compare to similar procedures described in the literature and applied by other researchers?
Author Response
Response to Reviewer 1
Point 1. The manuscript „Using Design of Experiments to Optimize a Screening Analytical Methodology Based on Solid-Phase Microextraction/Gas Chromatography for the Determination of Volatile Methylsiloxanes” in Molecules is interesting and it could be read by a wide group of scientists worldwide. However, it requires polishing in terms of presentation. It is rather chaotic in the current form.
Answer: The authors wish to express their appreciation to the Reviewer for the valuable comments on the manuscript. We hope that our revision can answer to the queries posed and may reflect an effective improvement of our work.
Point 2. Keywords: Design of Experiments; Volatile methylsiloxanes; Wastewater. Design of Experiments – why is this a keyword? Keywords need to be targeted so that scientists worldwide can use them for a literature search. "Validation" is a better choice of keyword, however still quite broad. Please add other keywords.
Answer: The authors have initially chosen 5 keywords - Design of Experiments; Volatile methylsiloxanes; Wastewater; Surface water; Solid-phase microextraction. We think that “Design of experiments” is in fact a keyword. Design of experiments involves not only the selection of suitable variables to the process, but also the planning of the experiments. It is a powerful statistical approach, involving data collection and analysis tools that can be used in a variety of experimental situations and whose use has been increasing in recent years. It allows for multiple input factors to be manipulated at the same time, identifying important interactions that may be missed when experimenting with one factor at a time. The authors agree that other important keyword may be “Validation” and also added “Gas Chromatography” as an extra keyword.
Point 3. How can the section Material and methods be contained within a subsection of Results and Discussion?? The research has to be conducted first, so that the Results can be described and interpreted in Discussion.
Answer: The authors understand the doubts of the reviewer. However, the "Material and Methods" section is not a subsection of "Results and Discussion", but an independent section. This structure was not a choice of the authors, but is defined by the journal. The authors only used the template provided by the journal, in which "Results and discussion" are presented in Section 2, while "Materials and methods" in Section 3.
Point 4. It would be best if there was a schematic drawing shown first, where it will be explained what will be done and how, in chronological order of method steps - e.g., the Ishikawa diagram. Then, add the tools which have been used for the purpose (including explanation and literature backing for their choice). Finally, please present the results obtained and their discussion (and conclude whether they are satisfactory or not). The order needs to be kept for clarity. It is interesting to attach also images of chromatograms.
Answer: Although the authors agree with the reviewer, the structure of the paper should follow the guidelines of the journal. For that reason, it is not possible to present the information as requested, which hampers the creation and the introduction of the suggested scheme. However, some chromatograms were added to the Supporting Information, as suggested.
Point 5. The information concerning the equipment used and the analytical procedure, including QA/QC, would be best presented in a table.
Answer: The authors searched several issues in this journal (and other journals of analytical chemistry) and found that this information is usually presented in the text form. Therefore, the authors suggest keeping the text in the current form.
Point 6. Conclusion - The Conclusion should summarize the conducted research in the context of already published data (existing knowledge). How does the described analytical procedure compare to similar procedures described in the literature and applied by other researchers?
Answer: The authors agree and changed the text.
Reviewer 2 Report
This report describes the optimization of a SPME GC-FID screen method for determination of volatile methylsiloxanes in water. While a similar methods (using SPME GCMS) have been reported previously for the determination of volatile methylsiloxanes, the application of Design of Experiments optimization and development of a screening method do appear to be original. The manuscript is generally well written. I offer some minor suggested changes below:
-In the abstract, the authors list the seven types of VMSs without explaining what they are. Perhaps these designations should be removed in the abstract.
-Line 74, replace with “These procedures are..”.
-Line 76 “techniques”
-Line 76 “determine” not “analyze”
-Line 92, “used as the screening”
-I think parts of section 2.1 and 2.2 would be better suited in the Material and Methods section.
-I am not very familiar with this type of optimization. Presumably it is more efficient that something like simplex optimization. Perhaps the advantages compared to other optimization methods should be discussed in more detail somewhere in the paper.
-Figure 3. Axes labels are a little too small and hard to ready.
-Table 4. I note that several compounds had low recoveries. Please discuss if the method is still acceptable for screening purposes. Presumably an accurate determination of the concentrations of these compounds would require standard additions calibration.
Author Response
Response to Reviewer 2
Point 1. This report describes the optimization of a SPME GC-FID screen method for determination of volatile methylsiloxanes in water. While a similar methods (using SPME GCMS) have been reported previously for the determination of volatile methylsiloxanes, the application of Design of Experiments optimization and development of a screening method do appear to be original. The manuscript is generally well written. I offer some minor suggested changes below.
Answer: The authors wish to express their appreciation to the Reviewer for the valuable comments on the manuscript. We hope that our revision can answer to the queries posed and may reflect an effective improvement of our work.
Point 2. In the abstract, the authors list the seven types of VMSs without explaining what they are. Perhaps these designations should be removed in the abstract.
Answer: They were removed from the abstract, as suggested.
Point 3. Line 74, replace with “These procedures are..”.
Answer: Done
Point 4. Line 76 “techniques”
Answer: Done
Point 5. Line 76 “determine” not “analyze”
Answer: Done
Point 6. Line 92, “used as the screening”
Answer: Done
Point 7. I think parts of section 2.1 and 2.2 would be better suited in the Material and Methods section.
Answer: The “Optimization of the extraction procedure” and “Method validation” are sections that describe and discuss experimental results. The materials and methods should describe the applied conditions (e.g., SPME, GC-FID, sample preparation, DoE, etc) to allow others to replicate and build on the published results. Therefore, the authors do not completely agree and would prefer to keep the information in the “Results and Discussion” section.
Point 8. I am not very familiar with this type of optimization. Presumably, it is more efficient that something like simplex optimization. Perhaps the advantages compared to other optimization methods should be discussed in more detail somewhere in the paper.
Answer: The definition of the optimal process conditions in analytical chemistry has been traditionally done using a one-factor-at-a-time approach. This methodology is, however, inefficient and time consuming, and it does not consider the interaction between variables.
The classical design of experiments approach to optimization (that includes the application of fractional factorial designs and central composite designs, for example) involves answering the following 3 questions: 1) What are the important factors? (Screening); 2) In what way do these factors affect the system? (Modelling); 3) What are the optimum levels? (Optimization). The resulting model can then be used to predict the treatment combination (experimental conditions) giving the optimum response. These systems are usually operated at some "threshold of acceptability". Simplex optimization is an alternative strategy, that asks the same questions but in reverse order (3 > 2 >1). Once in the region of the optimum, classical experimental designs can be used to model the system and determine factor importance in a limited region of the total factor space. The simplex is most powerful for continuous ("quantitative") variables. It can be used for discrete variables where there are several levels, and the levels can be logically ranked. It cannot be used for unranked discrete ("qualitative") variables. A sentence was introduced in the manuscript, highlighting the main advantages of the chosen optimization process.
Point 9. Figure 3. Axes labels are a little too small and hard to ready.
Answer: The figure has been enlarged, to improve the visualization of the axe labels. However, it was not possible to increase the font size since the specific software used to create those images did not allow it.
Point 10. Table 4. I note that several compounds had low recoveries. Please discuss if the method is still acceptable for screening purposes. Presumably, an accurate determination of the concentrations of these compounds would require standard additions calibration.
Answer: The authors considered that the recoveries obtained are quite acceptable for this type of screening analysis (higher than 70%) when compared to the literature. In fact, the lowest recovery obtained was 62 ± 8% for D5 in tap water. However, the authors also agree that in some conditions the use of a matrix-matched calibration should be an interesting alternative, which was already implemented for the analysis of wastewater samples, as described in the manuscript.
Reviewer 3 Report
The authors of the manuscript present a very interesting study in which a fully experimental design has been applied for the optimization of an analytical method to determine volatile methylsiloxanes in water. The method is based on head-space solid-phase microextraction followed by gas chromatography-flame ionization detection.
The manuscript is clear, well-written and organized. Thus, I strongly recommend the publication of this study in Molecules.
Some inputs to consider:
- Line 27: ‘…time, and 33 ºC…)
- Line 88: Specify the seven analytes.
- Line 209: Maybe it should be ‘relative recovery’
- Lines 266-267: are these mix solutions also prepared in acetone?
- Line 274: Specify that water samples are adjusted to 19.5%
- Line 285: According to Line 158, the desorption time should be 10 min
Author Response
Response to Reviewer 3
Point 1. The authors of the manuscript present a very interesting study in which a fully experimental design has been applied for the optimization of an analytical method to determine volatile methylsiloxanes in water. The method is based on head-space solid-phase microextraction followed by gas chromatography-flame ionization detection. The manuscript is clear, well-written and organized. Thus, I strongly recommend the publication of this study in Molecules.
Answer: The authors wish to express their appreciation to the Reviewer for the valuable comments on the manuscript. We hope that our revision can answer to the queries posed and may reflect an effective improvement of our work.
Point 2. Line 27: ‘…time, and 33 ºC…)
Answer: Done
Point 3. Line 88: Specify the seven analytes.
Answer: Done
Point 4. Line 209: Maybe it should be ‘relative recovery’
Answer: Done
Point 5. Lines 266-267: are these mix solutions also prepared in acetone?
Answer: Yes. This information was added to the text.
Point 6. Line 274: Specify that water samples are adjusted to 19.5%
Answer: This information was added to the text.
Point 7. Line 285: According to Line 158, the desorption time should be 10 min
Answer: The information was corrected.